# Two is better than one: Social rewards from two agents enhance offline improvements in motor skills more than single agent

**Masahiro Shiomi**[1]*, **Soto Okumura**[1,2], **Mitsuhiko Kimoto**[1,3], **Takamasa Iio**[1,4,5], **Katsunori Shimohara**[2]

**1** Department of Agent Interaction Design Laboratory, Advanced Telecommunications Research Institute International, Kyoto, Japan, **2** Department of Information Systems Design, Doshisha University, Kyoto, Japan, **3** Department of Information and Computer Science, Keio University, Kanagawa, Japan, **4** Social Cognitive Engineering Laboratory, University of Tsukuba, Ibaraki, Japan, **5** JST PRESTO, Japan

☯ These authors contributed equally to this work.
* m-shiomi@atr.jp

**Data Availability Statement:** All relevant data are within the manuscript and its Supporting Information files.

## Abstract

Social rewards as praise from others enhance offline improvements in human motor skills. Does praise from artificial beings, e.g., computer-graphics-based agents (displayed agents) and robots (collocated agents), also enhance offline improvements in motor skills as effectively as praise from humans? This paper answers this question via two subsequent days' experiment. We investigated the effect of the number of agents and their sense of presence toward offline improvement in motor skills because they are essential factors to change social effects and people's behaviors in human-agent and human-robot interaction. Our 96 participants performed a finger-tapping task. Our results showed that those who received praise from two agents showed significantly better offline motor skill improvement than people who were praised by just one agent and those who received no praise. However, we identified no significant effects related to the sense of presence.

## Introduction

Social rewards as praise from others not only promote positive emotional status but also physical changes. For example, past studies reported that praise provides various positive effects for humans: increased self-efficacy [1], improved motivation in children [2, 3], increased academic self-concept [4], greater pleasurable emotional reactions [5], and improved academic performances [6, 7]. Praise also positively influences physical changes, e.g., online improvement of task performances [8, 9]. Another previous study investigated praise effects on skill consolidation (i.e., offline improvement) in the offline consolidation of implicit sequence learning [9]. These results suggest that praise from others is a fundamental social reward [10].

Does praise from such artificial beings as computer-graphics-based agents (displayed agents) and robots (collocated agents) also positively affect humans? Past studies investigated the effect of praise from such agents. People perceived friendlier relationships with agents [11,

**Funding:** M.S. S.O. and K.S: JST CREST Grant, Number JPMJCR18A1, Japan M.K: JSPS KAKENHI Grant Numbers JP19J01290 T.I: JST PRESTO Grant Number JPMJPR1851, Japan, and JSPS KAKENHI Grant Numbers JP18H03311.

**Competing interests:** The authors have declared that no competing interests exist.

12], experienced improved motivation [13], and increased task online performances [14, 15]. A possible reason for such an effect of praise from such agents is that people regard them as social beings [16–18]. However, even if studies identified the effectiveness of praise, it remains unknown whether the praise from such agents has a similar effect on an offline consolidation, which is a critical learning process that is related to the amount of time spent on the learning [19, 20], as praise from humans. We hypothesize that praise from agents also directly influences the skill consolidation process. Our first research question: 1) Does praise from an agent influence the offline improvements of motor skills?

We are also interested in the factors that influence social rewards. Positive social feedback, which is an important factor in learning, improves offline learning. To increase the effects of such social feedback, the power of numbers is effective due to past human-human interaction literatures. For example, human behaviors and performance change consciously/unconsciously due to an increase in the number of people, such as social facilitation/loafing [21–25]. The power of the number effect is also observed in human-agent and human-robot interactions [26–28]. We hypothesize that the number of agents influences the effects of praise. Our second research question: 2) Does the number of agents increase the effects of the praise of offline improvements in motor skills?

Finally, we focused on the sense of the presence of the agents because researchers in human-robot interaction fields often experimentally investigated the advantages of collocated agents over displayed agents. Several studies claimed that collocated agents are perceived as more persuasive, lifelike, and positive [29–31]. Revealing the effectiveness of a sense of presence on praise effects would be useful knowledge for understanding how the characteristics of artificial agents influence human behaviors. We hypothesize that the sense of presence influences the effects of praise. Our third research question: 3) Does a sense of presence increase the effects of the praise of offline improvements on motor skills?

We verified these hypotheses on consecutive days in a two-day experiment that manipulated the timing of the praise, its number, the number of agents, and their sense of presence. Participants experienced training and praise (based on the conditions) on the first day and performed identical training on the second day without any information about the retest to minimize the possibility that they could practice the training task (Fig 1). These procedures enabled us to investigate the effects of praise on offline motor skill improvements.

This study is an extended version of a previous work [32] and contains additional conditions with more participants, analysis, and discussions. Moreover, this study design follows a past study that investigated praise effects toward the offline improvement from people [9]. We changed the form of the praise, i.e., agents praised participants instead of the human experimenter's movie.

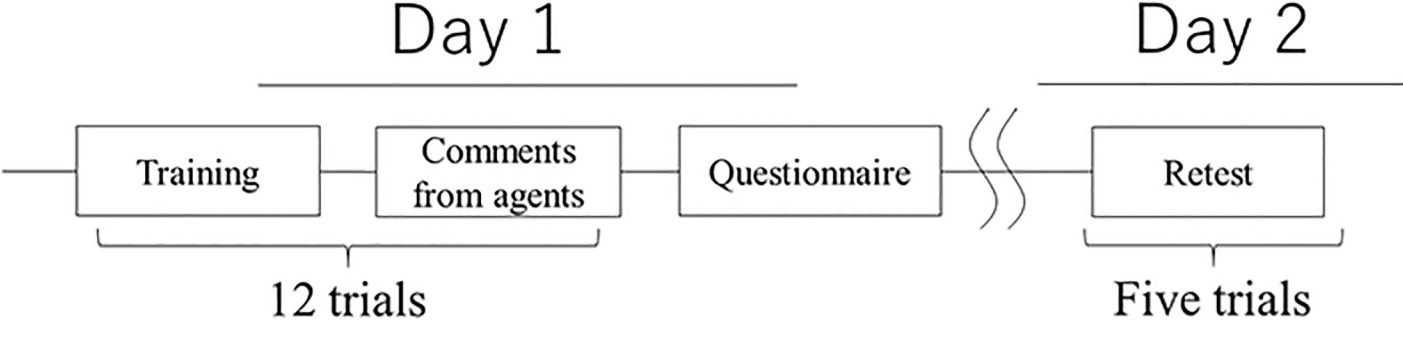

**Fig 1. Experiment design of study.**

## Material and method

### Participants

All 96 participants (48 women and 48 men who averaged 21.2 years of age, S.D: 1.37) gave informed consent for taking part in our two-day experiment. They were recruited by campus flyers at several universities in Japan and paid 500 Japanese yen (about $5). The ethics committee at the Advanced Telecommunication Research Institute approved this paper's methodology. None of the participants had any history of neurological, psychiatric, or sleep disorders.

### Environment

We conducted our experiment in a laboratory room at Doshisha University. Based on the condition, we installed either one or two displayed/collocated agents and a PC on a desk.

### Task design

We measured their offline motor skill improvements with a sequential, finger-tapping task, which has commonly been used in past studies, to focus on the effects of praise and offline improvements on motor tasks [9]. The participants repeatedly tapped four keys (1, 2, 3, 4) on a computer keyboard as quickly and as accurately as possible. Each trial period was 30 seconds. The sequence was identical for all participants and trials: 4, 1, 3, 2, and 4.

### Robot

In this study, we compared a collocated agent and a displayed agent. For the collocated agent, we used Sota, which was developed by VSTONE Inc. It has three degree of freedoms (DOFs) in its head, one DOF for each shoulder, one DOF for each elbow, and one for its base. It is 28 cm tall and weighs 763 g. It faced the participants and maintained a slight idling behavior to move its arms. For the displayed agent, we used Virtual-Sota, which is a computer-graphics-based agent that simulates Sota's motion and speech. Virtual-Sota expresses the same motions and sounds as Sota.

### Conditions

Our experiment used a 2 x 3 between-between-participant design, where participants were randomly assigned to one of two different senses of presence (displayed and collocated agents) and one of three different numbers of praise agents (no-praise, one-agent, and two-agent) by maintaining identical gender ratios among them. Thus, the participants were divided into six groups (Fig 2): "No-praise with one displayed agent," "Praise with one displayed agent," "Praise with two displayed agents," "No-praise with one collocated agent," "Praise with one collocated agent," and "Praise with two collocated agents."

**Presence factor.** We compared the task performance under two different presence conditions: displayed or collocated agent.

**Number factor.** We compared the task performance under three different number conditions. In the no-praise condition, we used one agent that always made neutral comments after each trial. In the one-agent condition, we also used one agent whose praise consisted of two sentences. In the two-agent condition, two agents used praise comments based on the rules; all the comments were separately used by two displayed/collocated agents.

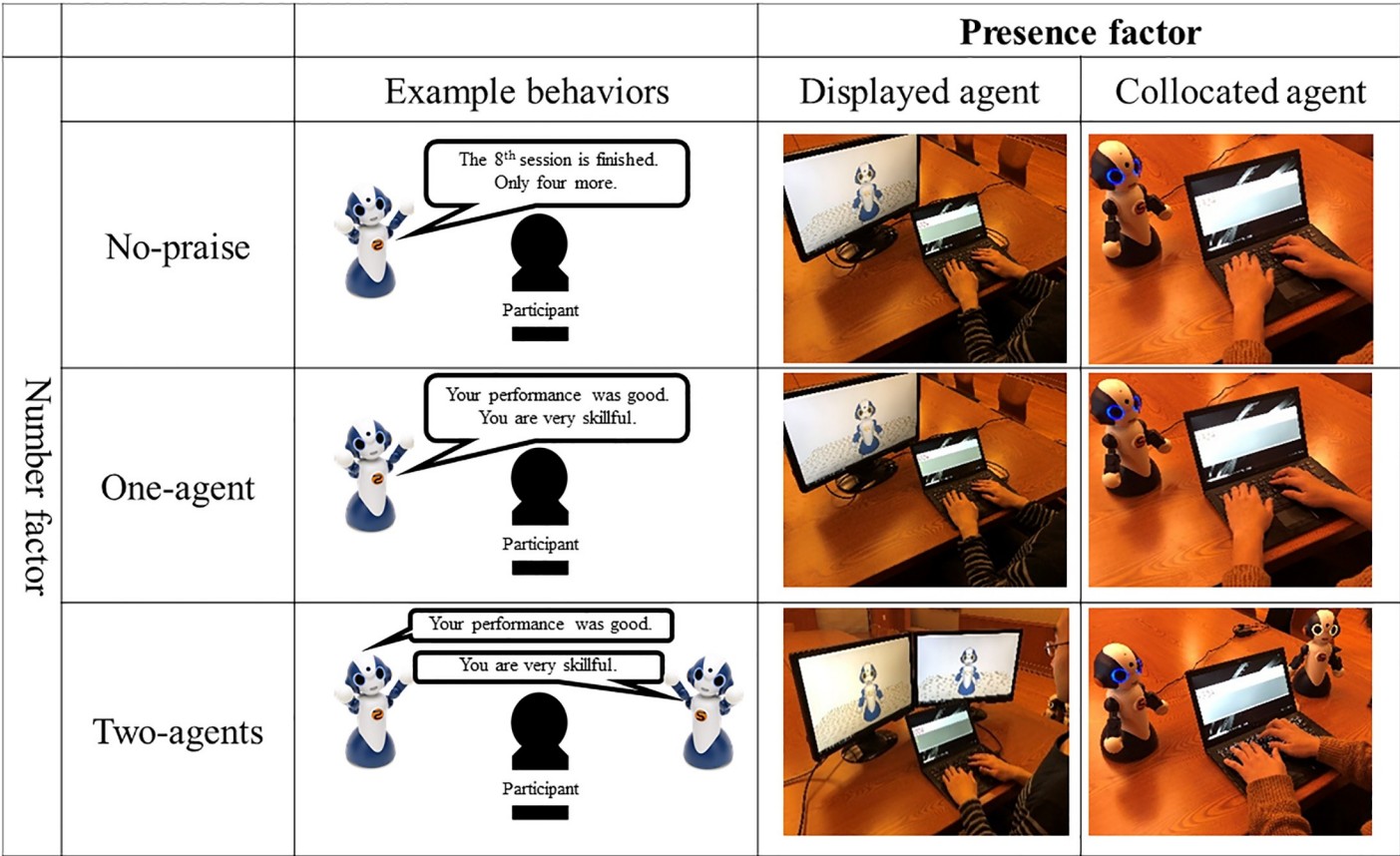

**Fig 2. Experiment design: In no-praise group, one displayed/collocated agent explains task's progress without any praise.** In one-agent group, one displayed/collocated agent gives praise during task. All praise contents include two sentences. In two-agent group, each agent gives one sentence of praise. Praise amounts are identical to one-agent group.

## Praise comments

By following the praise manipulation of past studies [9, 32], the agents provided comments to the participants after each trial. We prepared three kinds of comments to provide greater variety: pre-defined praise, neutral, and praise based on the task performance (Fig 3). Identical praise was not used on the same day.

The pre-defined praise included seven comments. Each comment consisted of two sentences, e.g., "your performance is very good, especially if this is your first trial" and "your typing speed seems faster than in the previous trial."

We had 12 neutral comments. We prepared ten comments about the current trial number and the remaining trials, e.g., "this completes the seventh trial, so you have five trials left." We also prepared two comments that consisted of the current trial number and the average performance, e.g., "this completes your third trial, and your performance resembles that of others."

For praise based on task performance, we implemented a simple rule-based algorithm to select praise comments (Fig 4) with three features: correct typed numbers, incorrect typed numbers, and total typed numbers. We prepared two comments for six categories: increase of correctly typed numbers, error-free typing, correct typing, decrease of typing errors, maintaining performance, and training attitude. The agents made these comments during the experiment to avoid skepticism based on overuse of the pre-defined praise comments, i.e., after the fourth and sixth trials. When the same category was selected by the algorithm for both timings,

| Order | Comment categories |
|:---:|:---:|
| 1 | Pre-defined praise comments |
| 2 | Pre-defined praise comments |
| 3 | Neutral comments |
| 4 | Praise comments based on their performance |
| 5 | Pre-defined praise comments |
| 6 | Praise comments based on their performance |
| 7 | Neutral comments |
| 8 | Pre-defined praise comments |
| 9 | Pre-defined praise comments |
| 10 | Pre-defined praise comments |
| 11 | Pre-defined praise comments |
| 12 | Pre-defined praise comments |

**Fig 3. Praise comments.**

different comments were used. The order of the praise categories is based on the priority of the algorithms (Fig 3B).

**Increase of correctly typed numbers.**   When the amount exceeds the maximum number at this point, the agents praised the number of correctly typed words: "since you correctly typed more numbers, your typing speed is increasing."

**Error-free typing.**   When a participant did not make any mistakes, the agents praised her accuracy: "since you made no mistakes, your typing is really accurate."

**Correct typing.**   When the participant's typing number exceeded a threshold (150 types in this study), the agents praised the typing amount: "your typing record is good since your number of correctly typed numbers is above average."

**Decrease of tying errors.**   When the amount of typing errors fell below a minimum number at this point, the agents praised: "since you've decreased your typing errors, it has become more accurate."

**Maintaining performance.**   When none of the above praise categories were selected and the total amount of typed numbers exceeded the last task, the agents praised the training task for maintaining its performance. Thus, the person's concentration was praised regardless of the performance: "you are maintaining your performance and concentrating."

**Training attitude.**   When the algorithm could not select any of the above categories, i.e., when the participant failed to maintain her performance or her amount of typing, the agents praised her attitude: "since your typing performance seems to have stabilized, your performance will probably increase."

```
//Pseudo code

if (correct_typing > before_correct_typing)
    praise (increasing_correct_typing)
else if (error_typing == 0)
    praise (no_miss)
else if (correct_typing > threshold_correct_typing)
    praise (certain_correct_miss)
else if (error_typing < before_error_typing)
    praise (decreasing_miss_typing)
else if (total_typing >= before_total_typing)
    praise (maintaining_performance)
else
    praise (attitude_for_training)
```

**Fig 4. Pseudo-code for praise comment control.**

## Procedure

All the participants came to our laboratory room on back-to-back days. On day 1, they did 12 trials of sequential finger-tapping tasks. The participants were told that the displayed/collocated agents would explain their roles and that the experimenter would explain the start/end timing of the tasks. After the experimenter left the room, the agents greeted the participants and explained their MC roles. The agents asked the participants to start the training task and explained that the experiment would end after all the trials were completed. The agents provided comments based on the conditions after each trial. The participants were given a 20-second break after each trial. At the end of day 1, the experimenter explained the next day's schedule (with a different task) to minimize the possibility that they would practice the training sequence before the evaluation of day 2.

On day 2, the participants experienced the same finger-tapping task in five trials (Fig 1A, right), different from the explanations of the last day. On day 2, the agents didn't praise; they just offered neutral comments.

## Measurement

We measured both objective and subjective items to evaluate the effects of praise. As objective items, we measured the performance of the last three training trials on day 1 and the first three retest trials on day 2. As subjective items, we gathered two questionnaire items to investigate the feeling of the perceived happiness to the agents' speech contents ("I felt happy after listening to the agent speeches"), and the degree of perceived praise ("I think that the agents praised me"). For these questionnaires, we used a response format on a seven-point-scale that described the options from 1 (strongly disagree) to 7 (strongly agree) [33].

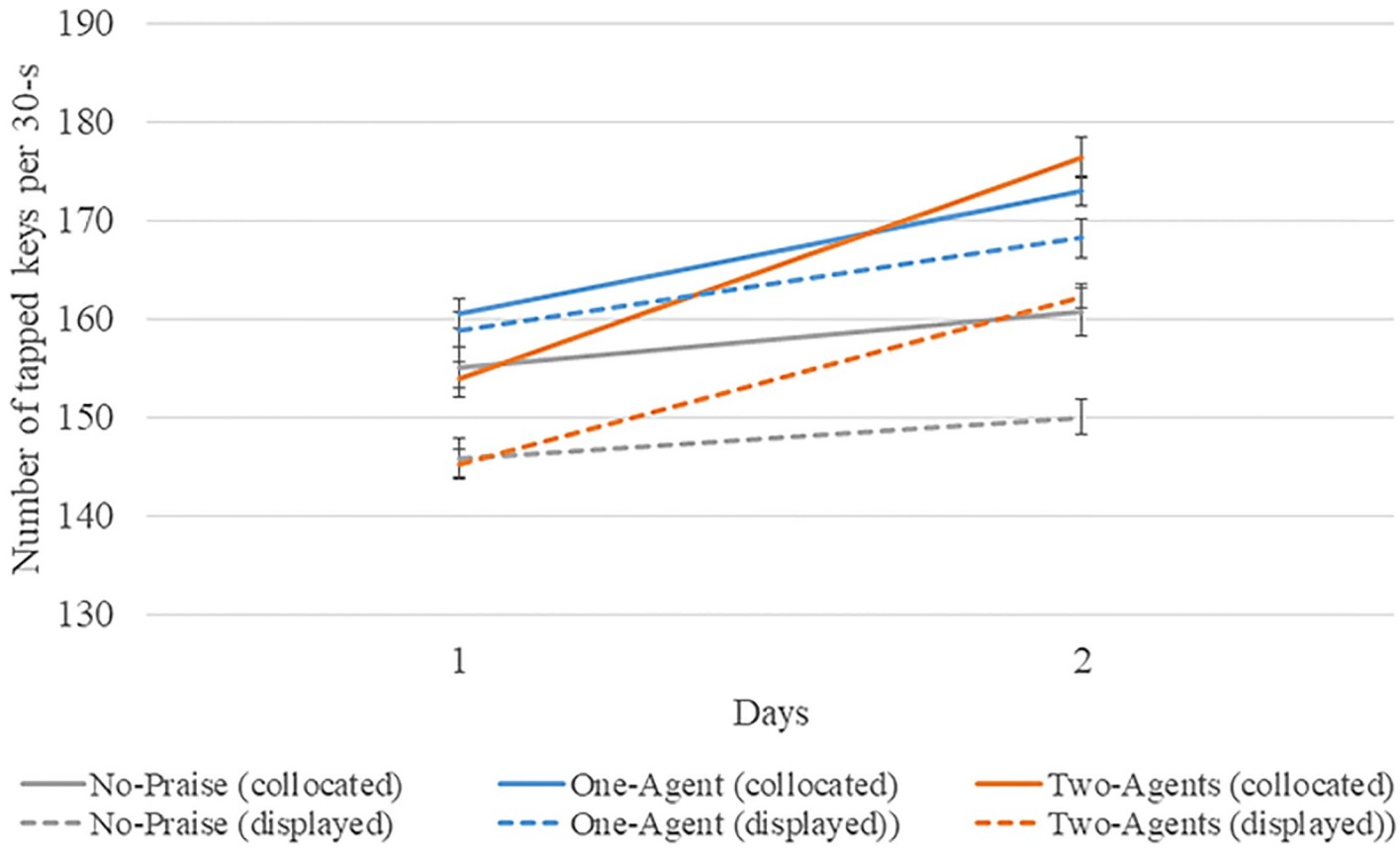

**Fig 5. Graph of mean performance of trained sequence (average and S.E.).**

## Results

### Performance of trained sequences

Fig 5 and Table 1 show the performance of the last three training trials on day 1 and the first three retest trials on day 2. First, we investigated the performances of each group at the end of day 1 as a manipulation check. A Levene's test indicated that the equality of the error variances was assumed at a significant level of 0.05 ($p = 0.840$), and a Kolmogorov-Smirnov test indicated that the data are consistent with a normal distribution ($p = 0.200$). A two-way analysis of variance (ANOVA) showed that the performance at the end of the training on day 1 did not show a significant difference for all the factors: the presence factor ($F(1, 90) = 1.197$, $p = .277$, *partial* $\eta^2 = .013$), the number factor ($F(2, 90) = 1.214$, $p = .302$, *partial* $\eta^2 = .026$), and the interaction effect ($F(2, 90) = 0.174$, $p = .840$, *partial* $\eta^2 = .004$). Therefore, the mean performances of the last three training trials on day 1 were not significantly different among the conditions.

Fig 6 shows the offline improvement of each condition, which is the percentage of increase from the mean performance of days 1 to 2. A Levene's test indicated that the equality of the error variances was assumed at a significant level of 0.05 ($p = 0.392$), and a Kolmogorov-Smirnov test indicated that the data are consistent with a normal distribution ($p = 0.200$). A two-way ANOVA showed a significant main effect in the number factor ($F(2, 90) = 29.4485$, $p < .001$, *partial* $\eta^2 = .396$). No significance was found in the presence factor ($F(1, 90) = 1.685$, $p = .198$, *partial* $\eta^2 = .018$) or the interaction effect ($F(2, 90) = 0.472$, $p = .625$, *partial* $\eta^2 = .010$).

**Table 1. Mean performance of trained sequence (average, S.D, and S.E.).**

| | | Day 1 | Day 2 | Day 1 | Day 2 | Day 1 | Day 2 |
|---|---|---|---|---|---|---|---|
| | | Average | | S.D. | | S.E | |
| | No-praise | 155.0 | 160.7 | 8.6 | 10.0 | 2.1 | 2.5 |
| Collocated | One-agent | 160.4 | 172.9 | 5.9 | 6.3 | 1.5 | 1.6 |
| | Two-agents | 153.8 | 176.3 | 6.9 | 8.3 | 1.7 | 2.1 |
| | No-praise | 145.8 | 150.0 | 8.0 | 7.4 | 2.0 | 1.9 |
| Displayed | One-agent | 158.9 | 168.1 | 7.5 | 8.1 | 1.9 | 2.0 |
| | Two-agents | 145.3 | 162.2 | 5.8 | 5.0 | 1.4 | 1.2 |

Multiple comparisons with the Bonferroni method revealed a significant difference: two > no-praise ($p < .001$), two > one ($p < .001$), and one > no-praise ($p = .021$). Therefore, praise from the displayed/collocated agents influenced the offline improvement of the motor skills. The number of agents significantly affected the offline improvement of the motor skills. On the other hand, even though a trend seemed to suggest the presence factor impacted the improvement, as shown in the graph, it did not significantly affect their offline improvements.

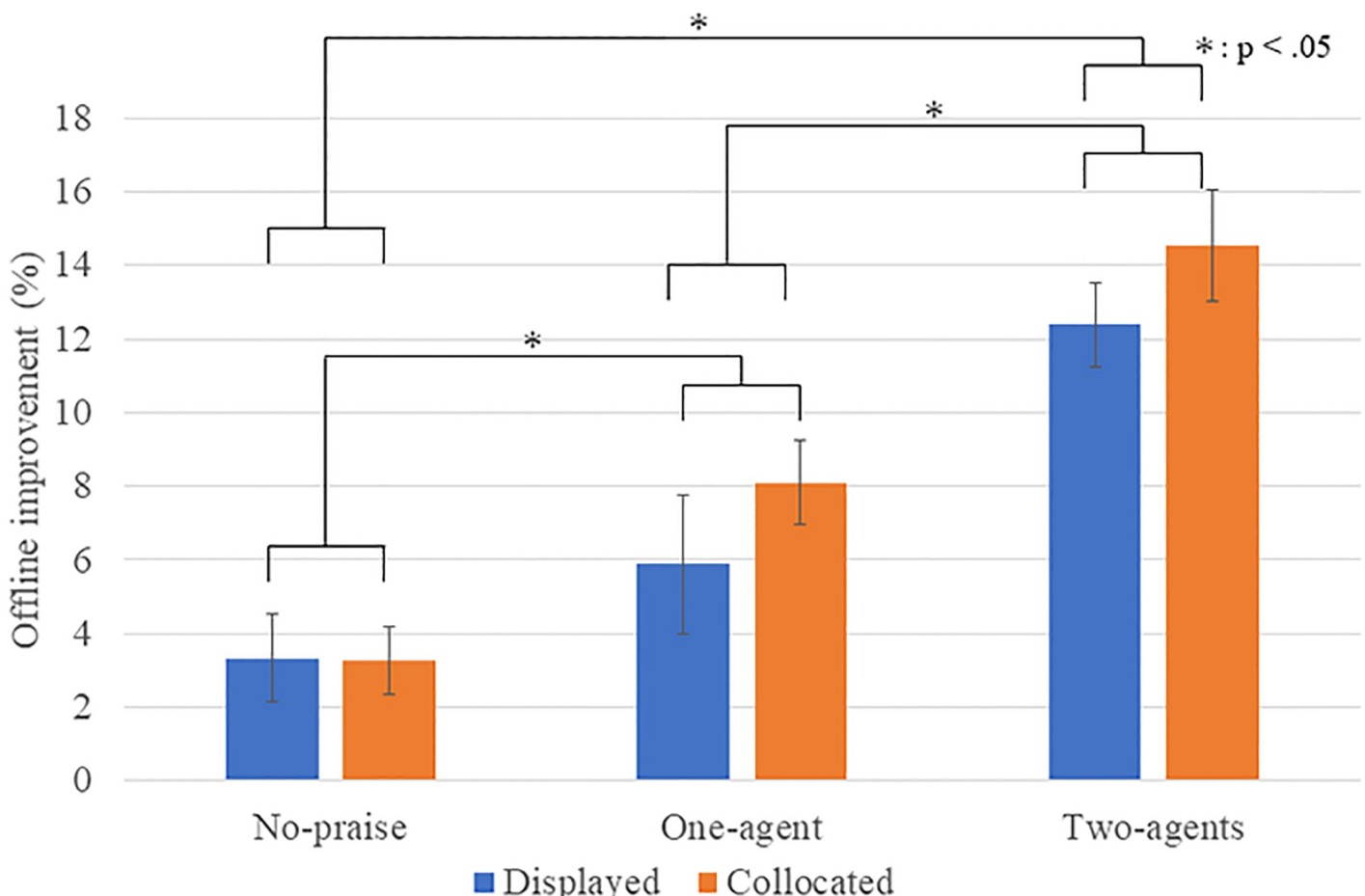

**Fig 6. Rate of offline improvement: Vertical axis indicates percentage of increase from mean performance of day 1 to 2.**

## Subjective impressions toward praise: Happiness rating and degree of perceived praise

At the end of day 1, we subjectively assessed the participant's happiness and degree of perceived praise by a questionnaire on a seven-point-scale (where 1 is most negative, and 7 is most positive). For the happiness rating (Fig 7), a Levene's test indicated that the equality of the error variances was assumed at a significant level of 0.05 ($p = 0.208$), but a Kolmogorov-Smirnov test did not indicate that the data are consistent with the normal distribution ($p<0.001$). Therefore, we performed a Box-Cox transformation for the questionnaire results. A two-way ANOVA showed a significant main effect in the number factor ($F(2, 90) = 25.312$, $p < .001$, partial $\eta^2 = .360$). No significance was found in the type factor ($F(1, 90) = 0.435$, $p = .511$, partial $\eta^2 = .005$) or the interaction effect ($F(2, 90) = 0.199$, $p = .820$ partial $\eta^2 = .004$). Multiple comparisons with the Bonferroni method revealed a significant difference: two > no-praise ($p < .001$) and one > no-praise ($p < .001$). We found no significant differences between two and one ($p = 1.000$).

Concerning the degree of perceived praise (Fig 8), a Levene's test indicated that the equality of the error variances was assumed at a significant level of 0.05 ($p = 0.029$), and a Kolmogorov-Smirnov test did not indicate that the data are consistent with the normal distribution ($p<0.001$). Therefore, we performed a Box-Cox transformation for the questionnaire results. A two-way ANOVA showed a significant main effect in the number factor ($F(2, 90) = 168.085$, $p < .001$, partial $\eta^2 = .789$). No significance was found in the type factor ($F(1, 90) = 0.584$, $p = .447$, partial $\eta^2 = .006$) or the interaction effect ($F(2, 90) = 0.578$, $p = .563$, partial $\eta^2 = .013$). Multiple comparisons with the Bonferroni method revealed a significant difference: two > no-praise ($p < .001$) and one > no-praise ($p < .001$). We found no significant differences between two and one ($p = 0.175$). Therefore, the results indicate successful manipulation of praise.

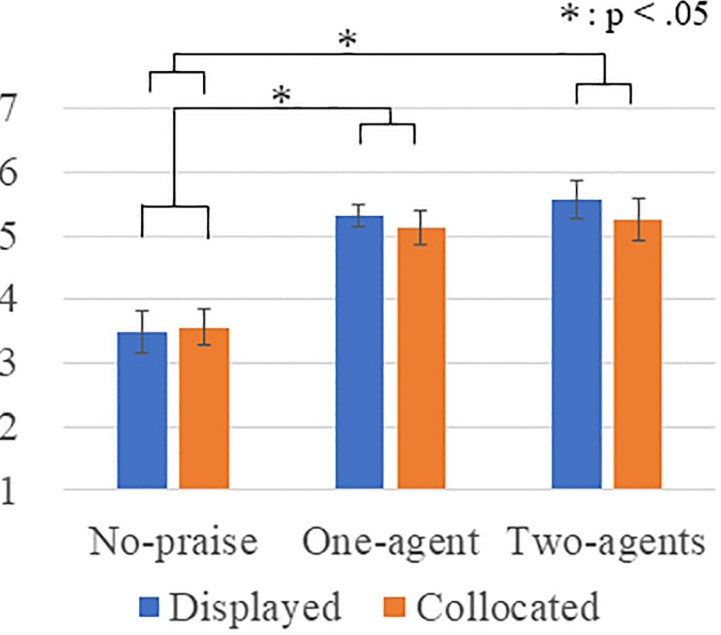

**Fig 7. Happiness rating.**

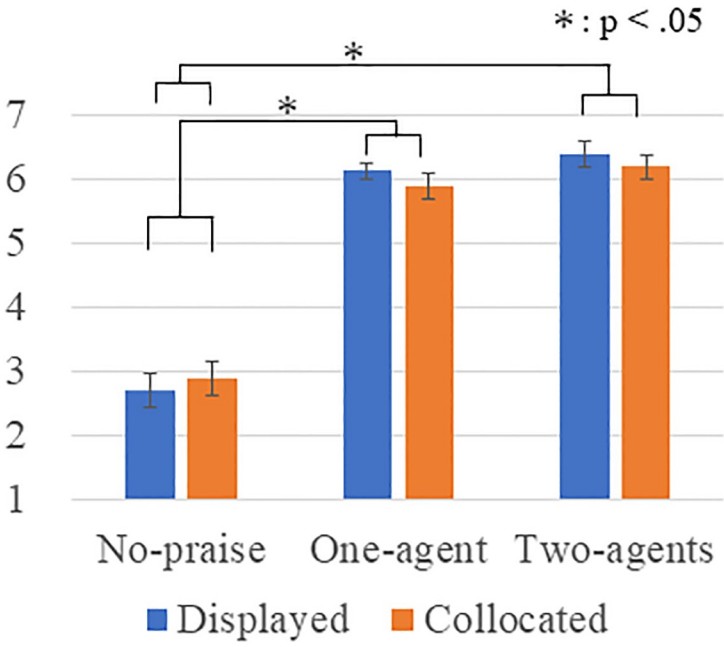

**Fig 8. Degree of perceived praise.**

## Discussion

### Design implications

In this study, we have three research questions, and our results showed support for R1 and R2, but not for R3. All the praised groups showed more offline motor skill improvements between days 1 and 2 than for the no-praise groups; the results showed the effectiveness of praise from artificial beings and humans [9, 19]. The number of praise agents enhanced the offline motor skill improvements, i.e., where praise from two agents was significantly better than praise from just one, even if the amount of praise was identical between the conditions.

Offline motor skill improvement would be useful for the education support of children, e.g., learning such basic computer skills as inputting characters by a keyboard. Moreover, if the effects of praise from multiple agents improve both offline motor skills and motivation, the knowledge will be helpful for designing educational support systems because such recent systems often use displaced/collocated physical agents to support learning. In fact, several studies investigated the effectiveness for such learning support agent systems [34, 35]. Since they reported some positive effects of praise from a single collocated robot, investigating praise effects on other education topics is interesting future work.

On the other hand, even though our experiment results suggest that the presence factor impacted the improvement, it did not significantly affect their offline improvements, i.e., the presence effects of agents is limited in our experimental contexts. A possible reason is the simplicity of the conversational interaction in our study. Past studies identified the effectiveness of such physical interaction as touching and hugging to change people's behaviors and their perceived feelings [36]; if collocated agents physically interact with participants, the praise effects might be changed. Of course the absence of evidence is not proof of an absence. Our results do not downplay the advantages of collocated agents compared to displayed agents.

Our experimental results pose another research question: does praise from two people enhance offline motor improvements more than praise from just one person? Since we focused

on the effects of artificial beings, we did not directly evaluate the effects of praise from multiple persons. Yet people regard displayed/collocated agents as social beings [16–18]. In fact, past studies and our study showed that the effect of praise from artificial agents resembles that from humans [21–25]. Thus, we assume that praise from two people would be more effective than praise from just one.

In summary, our study showed that praise from artificial beings improved skill consolidation in humans in a manner that resembles praise from human beings. The number of agents provides a basic effect for improving skill consolidation. Unlike past studies, our experimental results showed no significant effect of a sense of presence in the context of offline motor skill improvement.

### Effects of sense of presence, number, and environment

We compared the collocated and displayed agents to investigate the effects of praise from them. However, we are interacting with different types of agents, such as unembodied agents, voice-only agents, and simple-appearance collocated agents like a smart-speaker. Our experiment results did not show any significant effects in the sense of presence, although comparing these simpler agents would provide rich knowledge about praise effects.

In the context of the sense of presence, the number of agents is another essential factor to influence people's behaviors. Although this study showed that praise from two agents is better than one agent, we did not investigate the effects of three or more robots. The number of agents will increase such social influences and a saturation point will be identified. For example, past studies reported that the number of others influences social facilitation [37], social loafing [38], and peer pressure effects [39]. A past study on peer pressure effects from social robots reported that the number of robots changes peer pressure effects, but they also concluded that their total power is relatively weak compared to humans [40]. The praise effects, which have also been influenced by the number of robots, might be weaker than human participants. Investigating such effects is another interesting future work.

From another perspective, the differences of environments, i.e., physical and virtual reality, might be an important research topic. Several past studies investigated people's similar reactions to agents in VR/AR settings compared to collocated agents [41–43]. Our experiment results showed that the displayed agents showed relatively weaker effects than the collocated robots without any significant differences between them. But based on these related studies, immersive virtual reality or augmented reality will increase the effects of displayed agents. Therefore, investigating the praise effects from displayed agents between physical/virtual environments is another interesting future work.

## Conclusion

We investigated the effects of social rewards from agents for offline improvements in motor skills by considering the sense of presence (i.e., displayed and collocated agents) and the number of agents (one or two). We used a finger-tapping task and followed previous similar studies to investigate offline improvements and prepare praise comments for agents. Our experiment results showed that participants who received praise from two agents showed significantly better offline motor skill improvement than people who were praised by just one agent and those who received no praise. On the other hand, we identified no significant effects related to the sense of presence, unlike past studies that investigated its effects with displayed and collocated agents.

## Supporting information

**S1 File. Anonymized data set.**
(XLSX)

**S2 File.**
(PDF)

## Author Contributions

**Conceptualization:** Masahiro Shiomi, Mitsuhiko Kimoto, Takamasa Iio, Katsunori Shimohara.

**Data curation:** Masahiro Shiomi, Soto Okumura, Mitsuhiko Kimoto, Takamasa Iio.

**Formal analysis:** Masahiro Shiomi, Takamasa Iio.

**Investigation:** Masahiro Shiomi, Soto Okumura, Mitsuhiko Kimoto, Takamasa Iio, Katsunori Shimohara.

**Methodology:** Masahiro Shiomi, Soto Okumura, Mitsuhiko Kimoto, Takamasa Iio, Katsunori Shimohara.

**Project administration:** Masahiro Shiomi.

**Software:** Masahiro Shiomi.

**Supervision:** Masahiro Shiomi, Katsunori Shimohara.

**Validation:** Masahiro Shiomi, Mitsuhiko Kimoto.

**Writing – original draft:** Masahiro Shiomi, Soto Okumura, Mitsuhiko Kimoto, Takamasa Iio, Katsunori Shimohara.

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
