## [Decision Letter · Decision Letter 0]

6 May 2020

PONE-D-20-08172

Two is Better than One: Social Rewards from Two Agents Enhance Offline Improvements in Motor Skills More than Single Agent

PLOS ONE

Dear Dr. Shiomi,

Thank you for submitting your manuscript to PLOS ONE. After careful consideration, we feel that it has merit but does not fully meet PLOS ONE’s publication criteria as it currently stands. Therefore, we invite you to submit a revised version of the manuscript that addresses the points raised during the review process.

We would appreciate receiving your revised manuscript by Jun 20 2020 11:59PM. To enhance the reproducibility of your results, we recommend that if applicable you deposit your laboratory protocols in protocols.io, where a protocol can be assigned its own identifier (DOI) such that it can be cited independently in the future. For instructions see: http://journals.plos.org/plosone/s/submission-guidelines#loc-laboratory-protocols

We look forward to receiving your revised manuscript.

Kind regards,

Grzegorz Pochwatko, Ph.D.

Academic Editor

PLOS ONE

Journal Requirements:

2. Please include in your methods section a short description of how participants were recruited.

4. We noted in your submission details that a portion of your manuscript may have been presented or published elsewhere. Please clarify whether this [conference proceeding or publication] was peer-reviewed and formally published. If this work was previously peer-reviewed and published, in the cover letter please provide the reason that this work does not constitute dual publication and should be included in the current manuscript.

Additional Editor Comments (if provided):

The paper is sound and interesting, however it has to be improved for better clarity. Both reviewers have done excelent job in specifying what has to be done to reach perfect result. I strongly encourage You to introduce their suggestions and proceed

Reviewers' comments:

Reviewer's Responses to Questions

**Comments to the Author**

1. Is the manuscript technically sound, and do the data support the conclusions?

Reviewer #1: Partly

Reviewer #2: Yes

2. Has the statistical analysis been performed appropriately and rigorously? 

Reviewer #1: Yes

Reviewer #2: Yes

3. Have the authors made all data underlying the findings in their manuscript fully available?

Reviewer #1: No

Reviewer #2: No

4. Is the manuscript presented in an intelligible fashion and written in standard English?

Reviewer #1: No

Reviewer #2: Yes

5. Review Comments to the Author

Reviewer #1: The article introduces a study to show that positive feedback from virtual agents can improve offline learning. The study is timely and relevant for the field, although it suffers from a number of shortcommings that should be addressed. I detail them below.

English should be improved. For example, in the abstract:

Such social rewards as praise from others enhance offline improvements in our motor skills. Does praise from artificial beings, e.g., computer-graphics-based agents (virtual

should read:

Social rewards such as praise from others…

Also, in the introduction:

Thus, the following is our first research question:

should be rephrased.

Also:

The effects of enhancing offline improvement in motor skills resemble a kind of social influence from others. In the context of social influence, one leading factor in human-human interaction is the power of numbers; human behaviors and performance change consciously/unconsciously due to an increase of the number of people, such as social facilitation/loafing [19-23]. The power of the number effect is also observed in human-agent and humanrobot

interactions [24-26]. We hypothesize that the number of agents influences the effects of praise. Thus,

the following is our second research question:

This paragraph should be rewritten almost entirely to read better. For example:

The effects of enhancing offline improvement in motor skills resemble a kind of social influence from others.

You probably mean:

Positive Social feedback is an important factor in learning, and it improves offline learning.

In the disucssoin:

Still our study only…

This is again a strange construction.

I can find similar constructions all across the article, and would recommend addressing them by trying to put simpler grammatical structures, or have a native speaker thoroughly review the article.

Regarding references:

A key concept of the paper stated in the introduction is

offline consolidations (a critical process for learning)

but has no reference to point to. Although this is a basic concept, since this kind of article might also be read by software engineers with no background in learning science I would advise to add relevant references for this.

There are a striking number of similarities of this work with a work they cited (reference 9, which can be found here:

https://journals.plos.org/plosone/article?id=10.1371/journal.pone.0048174

Authors should better clarify inthe introduction that the designed is very similar (task, variable manipulated), but changing the way the praise is given (in one case a movie, in another a robot), and that the knowledge that the agent is not a real human (in the movie participants were told these were real people giving live feedback) does not change the main challenge.

Regarding experimental design:

The experiment is well designed, and it avoids a number of possible confounding factors.

however, in the no-physical condition actually rendered a character of similar size and position than the physical robot. It is possible to expect that an agent that is not-embodied (i.e., a voice with no body) would produce a different effect. I would therefore avoid the term non-physical, and rather opose a physical robot to a digital character.

Results.

I believe this could be improved.

Figure 2.a shows the average number of sequences completed in 30 seconds, but does not report on inter-subject variability which, I would argue, is quite important for a between group design. I would recomment including this.

Figure 2.b shows there is an effect of the number of agents. It also shows a trend in the impact of the physicality of the agent (a physical robot or a robot rendered on a screen).

However, the interpretation states clearly

authors should also explain what the vertical axis stands for in the caption of the figure.

authors should also state clearly that there seems to be a trend towards the fact that the physicality of the agent impacted the improvement, but the analysis method chosen (anova with bonferroni correction) does not reflect that.

It is entirely possible that the election of the analysis method is affecting this conclusion. I have not seen anywhere whether the measure shown in figure 2.b is normally distributed, which should be tested before performing an ANOVA analysis.

A linear regression, a generalized linear model or a bayesian analysis might be more suited than an ANOVA for this purpose.

In figure 2.c

rating questionnaires are ranked variables, and therefore not suited for anova analysis. Authors should resort to non-parametric tests, or do another kind of analysis.

I would also recommend providing the anonymized dataset together with the article submission in order to facilitate a re-analysis by other teams. If they provided the code

Discussion

I see a number of elements missing in the discussion that should be nuanced

Authors state: our experiment results did not show significant differences in physicality in the context of offline motor skill improvements

this can be misleading given the previous comments on the results of figure 2.b

-Immersive virtual reality (and recently, Immersive Augmented reality) has shown that the reaction to digital characters can be quite similar to physical agents, even with low quality of rendering. This is probably caused by the feeling of having a shared physical space with a digital agent, and therefore a social interaction much closer to real social interaction.

see, for example

https://dl.acm.org/doi/10.1145/1857893.1857896

https://journals.plos.org/plosone/article?id=10.1371/journal.pone.0032931

https://vhil.stanford.edu/pubs/2018/does-a-digital-assistant-need-a-body/

I recommend the authors to discuss this fact in the context of their experiment.

In addition to the nuancing of the impact of the physicality of the agent (and eventually, a re-analysis of that result), I would also recommend nuancing the statements regarding the number of agents. It is unlikely this effect would scale linearly. For example, whether having 4 robots, or 8 robots would improve the feedback (keeping the feedback sentences stable).

Structure.

In the results I can read this:

Each group had 16 participants, all of whom came to our laboratory

on back-to-back days. They were trained on a sequential finger-tapping task for which offline

improvement has previously been described [9, 30-32]. Their performance was defined by the number of

correctly tapped sequences per 30-second trials.

I believe this should be in the methods section.

I also can read this:

Note that the amount of praise is identical between groups; in the “Praise with one virtual agent” and “Praise with one physical agent” groups, the virtual/physical agents provided two sentences of praise. In the “Praise with two virtual agents” and “Praise with two physical agents” groups, each virtual/physical agent made just one sentence of praise.

This kind of clarification could probably be avoided if the methods section was before the results.

also, in the methods sectoin I found:

By following a past study’s praise manipulation

I would recommend that authors reference the study mentioned

Reviewer #2: PONE-D-20-08172

Two is Better than One: Social Rewards from Two Agents Enhance Offline Improvements in Motor Skills More than Single Agent

The paper presents one study aimed to determine whether the offline motor skills of a human partner could be improved by (1) the praise from a synthetic agent; (2) the number of agents that praise; (3) the type of physicality of the agent (virtual vs. physical).

The paper has many positive aspects (i.e., the proposed methodology is sound; the statistical treatments are adequate; Fig 1 and 2 are clear and very helpful to understand quickly the design and to picture the experiment done). The topic is relevant, and the paper could contribute significantly to the body of literature on Human Agent/Robot Interaction in general and specifically in an educational context.

However, there are some limitations (that can be fairly easily addressed) which should be addressed before publication.

Although the theoretical part is short, it is direct and explicit.

MISCELLANEOUS - MINOR ISSUES

The Fs, ps and other indicators should be in italics.

A table with the means and the standard deviations should be presented.

Some typos need to be corrected: ex: “popele would be effective moreat than praise from one person”

DOF = degree of freedom? Make it clear for the reader who is not used to with robotics.

MAJOR ISSUES

Although I do understand the initial intention of the authors, I would suggest a reorganization of the paper. Indeed, the present organization does not help to easily understand the experiment. For example, information about the experimental design, the procedure, or the material are either repeated in different sections (sometimes three times) or explained too “late” to have a big picture of the experiment that would help to understand the results (e.g., the reader has to wait the Results section to know that Happiness and Perceived Praise have been measured, as well as the way they were measured).

Consequently, I would suggest a more traditional organization to avoid repetitions: theoretical part, method with sample description, material description, procedure etc.. and then the results section and the discussion.

Moreover, how happiness and perceived praise were measured is not described. It is only said that it was measured “by a questionnaire on a seven-point-scale”. Instruments should be described as well as their reliability if applicable (e.g. Cronbach alpha).

Practical implications of the influence of praise on offline motor skills in educational context (for example) should be explored in the discussion.

6. PLOS authors have the option to publish the peer review history of their article (what does this mean?). If published, this will include your full peer review and any attached files.

Reviewer #1: Yes: Joan Llobera

Reviewer #2: No

---

## [Author Response · Author response to Decision Letter 0]

16 Jun 2020

We are grateful to the reviewers and editors for their detailed comments and specific feedback that enabled us to improve our paper. We considered all comments and tried to address them. We hope that the changes are satisfactory. In the attached document("Response to Reviewers" at the bottom of the PDF file), the comments from the reviewers are underlined.

---

## [Decision Letter · Decision Letter 1]

30 Sep 2020

Two is Better than One: Social Rewards from Two Agents Enhance Offline Improvements in Motor Skills More than Single Agent

PONE-D-20-08172R1

Dear Dr. Shiomi,

We’re pleased to inform you that your manuscript has been judged scientifically suitable for publication and will be formally accepted for publication once it meets all outstanding technical requirements.

Kind regards,

Grzegorz Pochwatko, Ph.D.

Academic Editor

PLOS ONE

Reviewers' comments:

Reviewer's Responses to Questions

**Comments to the Author**

1. If the authors have adequately addressed your comments raised in a previous round of review and you feel that this manuscript is now acceptable for publication, you may indicate that here to bypass the “Comments to the Author” section, enter your conflict of interest statement in the “Confidential to Editor” section, and submit your "Accept" recommendation.

Reviewer #1: All comments have been addressed

Reviewer #2: All comments have been addressed

2. Is the manuscript technically sound, and do the data support the conclusions?

Reviewer #1: Yes

Reviewer #2: Yes

3. Has the statistical analysis been performed appropriately and rigorously? 

Reviewer #1: Yes

Reviewer #2: Yes

4. Have the authors made all data underlying the findings in their manuscript fully available?

Reviewer #1: Yes

Reviewer #2: Yes

5. Is the manuscript presented in an intelligible fashion and written in standard English?

Reviewer #1: (No Response)

Reviewer #2: Yes

6. Review Comments to the Author

Reviewer #1: The comments have been addressed. Some English writing has been improved. The statistical analysis is quite more clear, now.

I think this work should be published

Reviewer #2: Authors have modified the main text according recommendations of both reviewers and responded to all the questions.

7. PLOS authors have the option to publish the peer review history of their article (what does this mean?). If published, this will include your full peer review and any attached files.

Reviewer #1: **Yes: **Joan Llobera

Reviewer #2: No

---

## [Editor Report · Acceptance letter]

8 Oct 2020

PONE-D-20-08172R1 

Two is Better than One: Social Rewards from Two Agents Enhance Offline Improvements in Motor Skills More than Single Agent 

Dear Dr. Shiomi:

I'm pleased to inform you that your manuscript has been deemed suitable for publication in PLOS ONE. Congratulations! Your manuscript is now with our production department. 

Kind regards, 

on behalf of

Dr. Grzegorz Pochwatko 

Academic Editor

PLOS ONE